# $f$-DIVERGENCE THERMODYNAMIC VARIATIONAL OBJECTIVE: A DEFORMED GEOMETRY PERSPECTIVE

## ABSTRACT

In this paper, we propose a $f$-divergence Thermodynamic Variational Objective ($f$-TVO). $f$-TVO generalizes the Thermodynamic Variational Objective (TVO) by replacing Kullback–Leibler (KL) divergence with arbitary differeitiable $f$-divergence. In particular, $f$-TVO approximates dual function of model evidence $f^*(p(x))$ rather than the log model evidence $\log p(x)$ in TVO. $f$-TVO is derived from a deformed $\chi$-geometry perspective. By defining $\chi$-exponential family exponential, we are able to integral $f$-TVO along the $\chi$-path, which is the deformed geodesic between variational posterior distribution and true posterior distribution. Optimizing scheme of $f$-TVO includes reparameterization trick and Monte Carlo approximation. Experiments on VAE and Bayesian neural network show that the proposed $f$-TVO performs better than cooresponding baseline $f$-divergence variational inference.

## 1 INTRODUCTION

Variational inference (VI) is a core technique in probabilistic machine learning (Mnih & Gregor, 2014; Mnih & Rezende, 2016; Tucker et al., 2017; Maddison et al., 2017; Paisley et al., 2012; Salimans et al., 2013; Ranganath et al., 2014; Kucukelbir et al., 2015). Normal Variational inference tries to perform a tractable posterior distribution approximation for log model evidence estimation $\log p(x)$. Commonly, this is done by introducing a divergence $D[q(z|x)||p(z|x)]$ between variational distribution $q(z|x)$ and true posterior $p(z|x)$ as a regularization term. Then the evidence lower bound (ELBO): $\log p(x) - D[q(z|x)||p(z|x)]$ is used as a surrogate optimization objective for maximum likelihood parameter estimation (Hoffman & Johnson, 2016; Duan et al., 2017; Yang, 2017).

Recently, there are serveral directions that generalize normal variational inference. Interested readers can refer to VI's background and progression in (Blei et al., 2017; Zhang et al., 2018). One direction is to explore tighter variational bounds (Burda et al., 2016; Chen et al., 2018; Masrani et al., 2019; Brekelmans et al., 2020). Among these efforts, the recent Thermodynamic Variational Objective (TVO) approach connects variational inference with thermodynamic integration. In paricular, TVO interprets the log model evidence estimation as 1D integral along the geodesic path between joint distribution of data and hidden variable $p(x, z)$ and variational hidden variable distribution $q(z|x)$. By properly choosing intergral intervals of Riemannian sum approximations, the TVO approximates the log likelihood with a series of upper and lower bounds, which are tighter surrogates than the normal variational inference objectives for model parameter likelihood maximization (Masrani et al., 2019).

Another direction is trying to extend the normal VI framework by replacing KL divergence with other statistical divergences. Among these efforts, Rényi's $\alpha$-divergence and $\chi$-divergence can be regarded as the root divergences, which makes many existing divergence as special cases including KL-divergence. Rényi's $\alpha$-VI (Li & Turner, 2016) and $\chi$-VI (Dieng et al., 2017) are stochastic VI that generalized from stochastic VI (Hoffman et al., 2013) and black-box VI (Ranganath et al., 2014), and outperform the classical KL-VI in Bayesian regressions and image reconstruction on some benchmarks. More generally, recent work $f$-divergence variational inference ($f$-VI) (Wan et al., 2020) gives a general framework that includes all divergences from $f$-divergence family. The variational objective of $f$-divergence VI can be regarded as the surrogate bound of dual function of model evidence $f^*(p(x))$.

| Divergences | $f(t)$ | $f^*(t)$ | Variational Bound |
|---|---|---|---|
| KL divergence | $t \log t$ | $-\log t$ | ELBO |
| KL divergence | $-\log t$ | $t \log t$ | EUBO |
| General $\chi^n$ divergence | $t^{1-n} - t, n \in \mathbb{R} \setminus \{0, 1\}$ | $t^n - 1$ | $\text{CUBO}_n$ |
| Total Varaition divergence | $|t - 1|$ | $|t - 1|$ | TVD |
| Rényi $\alpha$ divergence | $D_\alpha(q||p) = (\alpha - 1)^{-1} \log[1 + (\alpha - 1)\mathcal{H}_\alpha(q||p)]$ | | RVB |

Table 1: Difference divergences and their corresponding $f$ function. Rényi $\alpha$-divergence is not $f$-divergence, but it can be formulated with Hellinger divergence $\mathcal{H}_\alpha(q||p)$, which is an $f$-divergence.

In this paper, we combine the above two directions together by proposing $f$-TVO that generalizes the original TVO to arbitary $f$-divergence in a unified framework. $f$-TVO approximates dual function of model evidence $f^*(p(x))$ rather than the log model evidence $\log p(x)$ in TVO. The estimation of $f$-TVO is established from a deformed $\chi$-geometry prospective. By defining $\chi$-exponential family distribution, the computation of $f$-TVO is integrated over the $\chi$-path, which is the geodesic between $p(x, z)$ and $q(z|x)$ under $\chi$-geometry. The proposed $f$-TVO is a tighter bound of dual function of model evidence $f^*(p(x))$ rather than the normal variational lower bound in TVO.

Our contributions include the following:

- We enrich the TVO by proposing a unified f-TVO framework that compatible with an amount of existing divergence, such as KL, $\alpha$ divergence, $\chi$ divergence;
- We derive a unified f-TVO bound equiped with the upper/lower bound criteria that improves many existing bounds, such as ELBO, CUBO, RVB;
- Experiments show that our proposed f-TVO is a better optimization surrogate than corresponding f-VI.

The rest of the paper is organized as follows: Sec. 2 reviews preliminary knowledges for $f$-divergence and Thermodynamic Variational Objective, Sec. 3 presents our $f$-TVO, Sec. 4 extends our approach to more variants of $f$-TVO, and Sec. 5 presents our experiment results.

## 2 PRELIMINARY

In this section, we will review the preliminary knowledge for $f$-divergence and Thermodynamic Variational Objective.

### 2.1 $f$-DIVERGENCE

$f$-divergence is a measures of difference between two probability distributions. It can be defined as follows (Sason & Verdú, 2016).

**Definition 1.** *Given a convex function $f$ with $f(1) = 0$, then $f$-**divergence** from probability distribution $q(z)$ to $p(z)$ is defined as:*

$$D_f(p||q) = \int f(\frac{q(z)}{p(z)})p(z)dz = \mathbb{E}_p[f(\frac{q(z)}{p(z)})] \qquad (1)$$

Many existing well-know divergence can be regarded as special cases of $f$-divergence by properly choose the generation function $f$. Table 1 shows the relationship between the sample divergences and their corresponding generation function $f$.

Wan *et al.*(Wan et al., 2020) proposed a $f$-divergence variational inference ($f$-VI) that generalizes normal variational inference to all f-divergences in a unified framework. $f$-VI framework is primarily based on reverse $f$-divergence (Wan et al., 2020). One can connect forward $f$-divergence $D_f(p||q)$ with the reverse $f$-divergence $D_f(q||p)$ via dual function $f^*$ as follows.

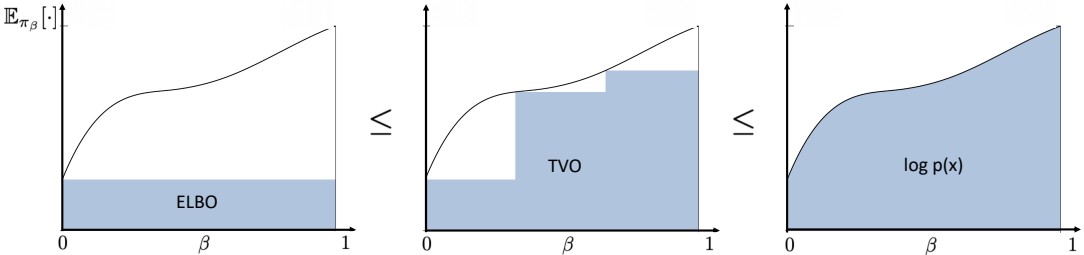

Figure 1: Thermodynamic Variational Objective (TVO). Left:evidence lower bound (ELBO); middle: TVO is a Riemannian sum approximation of $\log p(x)$; right: log model evidence $\log p(x)$. TVO is a tighter bound of $\log p(x)$ than ELBO.

**Definition 2.** *Given a function f, its* **dual funtion**[1] $f^*$ *is defined as:*

$$f^*(t) = t \cdot f(1/t) \tag{2}$$

The dual function $f^*$ has the following properties: $(1)(f^*)^* = f$, $(2)f^*$ is convex iff $f$ is convex, $(3)f^*(1) = 0$ iff $f(1) = 0$. $f$-divergence in equation 1 can be reformulated with the dual function $f^*$:

$$D_f(p||q) = \int f(\frac{q(z)}{p(z)})p(z)dz = \int f^*(\frac{p(z)}{q(z)})q(z)dz = D_{f^*}(q||p) \tag{3}$$

As shown in the theorem below, $f$-variational bound is an approximation of the dual function $f^*$ of the model evidence $p(x)$:

**Theorem 1.** *The dual function of model evidence $f^*(p(x))$ is bounded by the $f$-variational bound $\mathcal{L}_f(q, x)$*

$$\mathcal{L}_f(q, x) = \mathbb{E}_{q(z|x)}[f^*(\frac{p(z, x)}{q(z|x)})] \geq f^*(p(x)) \tag{4}$$

*where equality is obtained when $q(z|x) = p(z|x)$.*

The proof of the theorem is straightforward as $f^*$ is convex (Wan et al., 2020).

Many well-known variation bounds can be represented in the $f$-VI framework. For example, $f$-VI recovers to ELBO when $f = t \log t$, and $f$-VI recovers to CUBO$_n$ when $f = t^{1-n} - t$. More details are shown in the Table 1.

## 2.2 THERMODYNAMIC VARIATIONAL OBJECTIVE

The evidence lower bound (ELBO) is a lower bound of the log evidence of a generative model $p(x, z)$. It can be formulated as the difference between log evidence of model evidence $\log p(x)$ and KL divergence of variational posterior distribution $q(z|x)$ and true posterior distribution $p(z|x)$:

$$\text{ELBO} = \log p(x) - D_{KL}(q(z|x)||p(z|x)) \tag{5}$$

The Thermodynamic Variational Objective (TVO) gives a tighter bound of $\log p(x)$ than ELBO. As shown in Fig. 1, TVO is a Riemannian sum of integral along the geometric path between the model and inference network:

$$\underbrace{\frac{1}{K}\Big(\text{ELBO} + \sum_{k=1}^{K-1} \mathbb{E}_{\pi_{\beta_k}}[\log \frac{p(x, z)}{q(z|x)}]\Big)}_{\text{TVO}} \leq \int_0^1 \mathbb{E}_{\pi_\beta}[\log \frac{p(x, z)}{q(z|x)}]d\beta = \log p(x) \tag{6}$$

---

[1]Note:dual function here is different from Fenchel conjugate function: $f^c(y) = \sup_{x \in \text{dom } f}(y^\top x - f(x))$.

where $\beta = [\beta_0, \beta_1, \cdots, \beta_K]$ is a partition between [0,1], where $\beta_0 = 0$, and $\beta_K = 1$, $\pi_\beta$ is the normalized geometric combination of $p(x,z)$ and $q(z|x)$, i.e. $\tilde{\pi}_\beta = q(z|x)^{1-\beta} p(x,z)^\beta$ and $\pi_\beta = \tilde{\pi}_\beta / Z_\beta$, where $Z_\beta$ is a normalization constant for each $\beta$. One can verify that $\pi_\beta$ is a member of exponential family distribution, no matter what $q(z|x)$ and $p(x,z)$ are:

$$\pi_\beta(z|x) = \pi_0(z|x) \exp\{\beta \cdot T(x,z) - \phi(x;\beta)\} \tag{7}$$

where $T(x,z) = \log \frac{p(x,z)}{q(z|x)}$, $\phi(x;\beta)$ is the log partition function that normalize over $z$.

## 3 $\chi$-THERMODYNAMIC VARIATIONAL OBJECTIVE

In this section, we establish our theory for $f$-Thermodynamic Variational Objective ($f$-TVO). We first connect $f$-divergence with $\chi$-geometry. Then, we present $f$-TVO as an integration along the $\chi$-path. Lastly, we specify the computation process of $f$-TVO and its gradients.

### 3.1 $\chi$-LOGARITHM AND $\chi$-EXPONENTIAL

We first define a generalized $\chi$-logarithm function for a positive non-increasing function $\chi$:

$$\log_\chi(u) = \int_1^u \chi(v) dv \tag{8}$$

which is different from the definition in (Naudts, 2011; Amari, 2016). Notice that equation 8 gives normal log function $\log u$ when $\chi(v) = \frac{1}{v}$. The inverse of the $\chi$-logarithm is called $\chi$-exponential:

$$\exp_\chi(u) = 1 + \int_0^u \psi(v) dv \tag{9}$$

where $\psi$ is an auxiliary function. One can determine $\psi$ for each $\log_\chi(u)$, e.g. $\psi(v) = \exp(v)$ when $\log_\chi(u) = \log u$. $\chi$-Logarithm and $\chi$-Exponential have the following properties:

$$(1) \; \frac{d}{du}\exp_\chi(u) = \psi(u), \quad (2) \; u = \exp_\chi(\log_\chi(u)), \quad (3) \; \frac{1}{\chi(u)} = \psi(\log_\chi(u)).$$

For any divergence with convex diffrentiable generation function $f$ with $f(0) = 1$, there exists a continuous positive non-increasing function $\chi$ that connects the dual function $f^*$:

$$f^*(u) = -\log_\chi(u) = -\int_1^u \chi(v) dv \tag{10}$$

then $f$-divergence can be represented as

$$D_f(q||p) = D_{f^*}(p||q) = -\int p(z) \cdot \log_\chi(\frac{q(z)}{p(z)}) dz \tag{11}$$

### 3.2 THERMODYNAMIC INTEGRATION

Using the $\chi$-logarithm function and $\chi$-exponential function defined in equation 8 and equation 9, we can define a exponential family distribution under $\chi$-geometry:

**Definition 3.** $\chi$-*Exponential Family Distribution*:

$$\log_\chi \frac{p(\mathbf{x}, \theta)}{c(x)} = \theta \cdot \mathbf{x} - \Phi(\theta) \tag{12}$$

*or equivalently:*

$$p(\mathbf{x}, \theta) = c(x) \exp_\chi\{\theta \cdot \mathbf{x} - \Phi(\theta)\} \tag{13}$$

*where $x$ is random variable, $\theta$ is parameter, $\Phi(\theta)$ is the partition function, $c(x)$ is the normalization term.*

Given two unnormalized distribution $\tilde{\pi}_0(z)$ and $\tilde{\pi}_1(z)$, we define an unnormalized distribution that is the log $\chi$ interpolation between $\tilde{\pi}_0(z)$ and $\tilde{\pi}_1(z)$:

$$\log_\chi(\tilde{\pi}_\beta(z)) = (1 - \beta)\log_\chi(\tilde{\pi}_0(z)) + \beta\log_\chi(\tilde{\pi}_1(z)) \qquad (14)$$

Its normalized distribution is defined as:

$$\pi_\beta(z) = \frac{\tilde{\pi}_\beta(z)}{Z_\beta} \qquad (15)$$

where $Z_\beta = \int \tilde{\pi}_\beta(z)dz$. It's easy to show that: $\tilde{\pi}_\beta(z)$ is on the log $\chi$ geodesic between $\tilde{\pi}_0(z)$ and $\tilde{\pi}_1(z)$ with $\chi$-Exponential Family Distribution as:

**Theorem 2.** $\pi_\beta$ is also an $\chi$-exponential family distribution, no matter $\tilde{\pi}_0(z)$ and $\tilde{\pi}_1(z)$.

*Proof.*

$$\begin{aligned}\tilde{\pi}_\beta(z) &= \exp_\chi((1 - \beta)\log_\chi \tilde{\pi}_0(z) + \beta\log_\chi \tilde{\pi}_1(z)) \\ &= \exp_\chi(\log_\chi \tilde{\pi}_0(z) + \beta(\log_\chi \tilde{\pi}_1(z) - \log_\chi \tilde{\pi}_0(z)))\end{aligned} \qquad (16)$$

$\square$

By defining the potential energy function: $U_\beta(z) = \log_\chi \tilde{\pi}_\beta(z)$, then we can estimate the $\log_\chi$ of the ratio of the normalizing constants via thermodynamic integration:

$$\log_\chi Z_1 - \log_\chi Z_0 = \int_0^1 \int \frac{\chi(Z_\beta)}{\chi(\tilde{\pi}_\beta(z))}(\log_\chi \tilde{\pi}_0(z) - \log_\chi \tilde{\pi}_1(z))dzd\beta \qquad (17)$$

Notice that variational inference can be connected with thermodynamic integration if we set:

$$\tilde{\pi}_0(z) = q(z|x), \quad Z_0 = \int q(z|x)dz = 1$$

$$\tilde{\pi}_1(z) = p(x,z), \quad Z_1 = \int p(x,z)dz = p(x) \qquad (18)$$

then the $\log_\chi$ of model evidence is the thermodynamic integration (TI):

$$\log_\chi p(x) = \int_0^1 S_\beta d\beta \qquad (19)$$

where $S_\beta = \int \frac{\chi(Z_\beta)}{\chi(\tilde{\pi}_\beta(z))}(\log_\chi q(z|x) - \log_\chi p(x,z))dz$

Recall the relation between $f$-divergence and $\chi$ function in equation 10, then $\log_\chi p(x) = f^*(p(x))$ is dual function $f$ of the model evidence, which corresponds to the $f$-VI in (Wan et al., 2020).

To approximate $f$-TVO defined in equation 17, we pick the a list of ordered partitions $\beta = [\beta_0, \beta_1, \cdots, \beta_K]$, where $\beta_0 \leq \beta_1 \leq \cdots \leq \beta_K$, $\beta_0 = 0, \beta_K = 1$, then our $f$-TVO is the left Riemannian approximation of TI:

$$f\text{-TVO}^L(K) = \sum_{k=1}^K \Delta_k S_{\beta_{k-1}}, \qquad (20)$$

where $\Delta_k = \beta_k - \beta_{k-1}$.

### 3.3 COMPUTATION

In this section, we show how to efficiently compute the $f$-TVO.

We first rewrite $S_\beta$ in equation 17 into

$$\begin{aligned}S_\beta &= \int \frac{\chi(Z_\beta)}{\chi(\tilde{\pi}_\beta(z))}(\log_\chi \tilde{\pi}_0(z) - \log_\chi \tilde{\pi}_1(z))dz \\ &= \mathbb{E}_{q(z)}[\frac{\chi(Z_\beta)}{\chi(\tilde{\pi}_\beta(z))}(\log_\chi \tilde{\pi}_0(z) - \log_\chi \tilde{\pi}_1(z))/q(z|x)]\end{aligned} \qquad (21)$$

We apply the reparameterization trick to estimate $S_\beta$. We assume there is a mapping $g_\phi(\epsilon)$ that satisfies: $z = g_\phi(\epsilon)$. Then the expectation of arbitrary function $F(z)$ over distribution $q(z)$ can be computed as $\mathbb{E}_{q(z)}[F(z)] = \mathbb{E}_\epsilon[F(g_\phi(\epsilon))]$. In particular, we use the prevalent Gaussian reparameterization: $z \sim \mathcal{N}(\mu, \Sigma)$, i.e. $z = \mu + \Sigma^{\frac{1}{2}}\epsilon, \epsilon \sim \mathcal{N}(0, I)$. Then $S_\beta$ can be reformulated as:

$$S_\beta = \mathbb{E}_\epsilon\Big[\frac{\chi(Z_\beta)}{p(\epsilon)\chi(\tilde{\pi}_\beta(g_\phi(\epsilon)))}(\log_\chi p(\epsilon) - \log_\chi p(x, g_\phi(\epsilon)))\Big] \tag{22}$$

where we define $q(z|x) = p(\epsilon)$ and $p(x, z) = p(x, g_\phi(\epsilon))$ in equation 19. Empirically, we can compute $S_\beta$ as:

$$\hat{S}_\beta = \frac{1}{N}\sum_{n=1}^{N}[w_n(\log_\chi p(\epsilon_n) - \log_\chi p(x, g_\phi(\epsilon_n)))] \tag{23}$$

where $w_n = \chi(\sum_{n=1}^{N}\tilde{\pi}_\beta(g_\phi(\epsilon_n)))/p(\epsilon_n)\chi(\tilde{\pi}_\beta(g_\phi(\epsilon_n)))$.

In practice, we split the computation of $\hat{S}_\beta$ into two parts: $\hat{S}_\beta = L - R$:

$$L = \frac{1}{N}\sum_{n=1}^{N}[w_n\log_\chi p(\epsilon_n)]$$

$$R = \frac{1}{N}\sum_{n=1}^{N}[w_n\log_\chi p(x, g_\phi(\epsilon_n))] \tag{24}$$

This is because we use $\log p(x, g_\phi(\epsilon_n))$ and $\log p(\epsilon_n))$ during our computation to avoid numerical overflow or underflow. All the intermediate sum operations are performed by *logsumexp* operation until we get $\log L$ and $\log R$, then we use exponential operation to complete the computation.

## 4 GENERALIZING THERMODYNAMIC OBJECTIVES

Recall $f$-TVO in equation 20 is left Riemannian approximation of Thermodynamic integration. Similarly, We can also have right Riemannian approximation of $f$-TI:

$$f\text{-TVO}^R(K) = \sum_{k=1}^{K}\Delta_k S_{\beta_k} \tag{25}$$

Both left and right Riemannian approximation corresponds to existing variational bounds. For example, left single Riemannian approximation of the KL-TVO equals the ELBO, while the right single Riemannian approximation equals to the EUBO. Also, it is easy to verify if the dual function $f^*$ is increasing (decreasing), then left Riemannian approximation is an upper (lower) bound of Thermodynamic integration, while right Riemannian approximation is a lower (upper) bound of Thermodynamic integration.

In addition to the left or right Riemannian approximation of Thermodynamic integration, we can also define a "zig-zag" (z) approximation of Thermodynamic integration, which combines left and right Riemannian approximation. As shown in FIg. 2, the "zig-zag" Thermodynamic objective can be formulated as:

$$f\text{-TVO}^Z = \sum_{k=1}^{K}\Delta_k[\mathbf{1}(k\%2 = 1)S_{\beta_{k-1}} + \mathbf{1}(k\%2 = 0)S_{\beta_k}] \tag{26}$$

where $\mathbf{1}(A)$ is indicator function, i.e. $\mathbf{1}(A) = 1$ when condition $A$ is true, $\mathbf{1}(A) = 0$ when condition $A$ is false.

## 5 EXPERIMENTS

In this section, we evaluate the effectiveness and the wide applicability of $f$-TVO on two tasks. $f$-TVO is first used as a surrogate optimization objective of $f$-VI in a Bayesian neural network for linear regression, then used in a VAE for image reconstruction and generation. For both tasks, we use Adam as our optimizer with recommended parameters in (Da, 2014). For all experiments, we use even partition of the temperature for Riemannian integration.

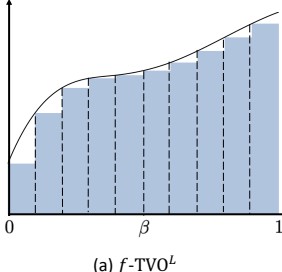 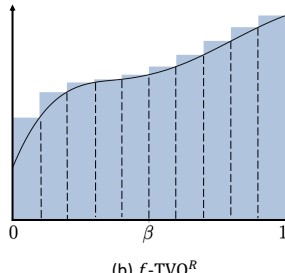 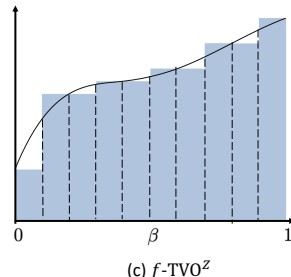

(a) $f$-TVO$^L$  (b) $f$-TVO$^R$  (c) $f$-TVO$^Z$

Figure 2: Variants of $f$-Thermodynamic Variational Objective ($f$-TVO). (a)Left Riemannian approximation; (b) Right Riemannian approximation; (c)zig-zag Riemannian approximation.

| Dataset | KL-VI | KL-TVO | $\chi$-VI | $\chi$-TVO | $\alpha$-VI | $\alpha$-TVO | $f_{c1}$-VI | $f_{c1}$-TVO |
|---|---|---|---|---|---|---|---|---|
| Caltech 101 | 73.72 | **73.51** | 73.84 | 73.61 | 74.95 | 74.23 | 74.90 | 74.62 |
| Frey Face | 160.85 | 160.58 | 160.61 | **160.32** | 161.06 | 160.85 | 160.73 | 160.35 |
| MNIST | 59.03 | **58.89** | 62.15 | 61.77 | 61.88 | 61.42 | 59.47 | 59.21 |
| Omniglot | 109.65 | 109.52 | 110.57 | 109.89 | 110.75 | 110.12 | 108.31 | **108.09** |

Table 2: Test reconstruction errors of $f$-TVO VAEs. Lower is better. In most cases, $f$-TVO outperform corresponding $f$-VI.

### 5.1 BAYESIAN VARIATIONAL AUTOENCODER

$f$-TVO is used in Bayesian VAE for image reconstruction task on datasets including Caltech 101 Silhouettes (Cal), Frey Face (Fre), MNIST (MNI), and Omniglot (Omn). We have replaced the conventional ELBO loss function of VAE with the proposed $f$-TVOs that correspond to flexible $f$-divergences. We test and compare the $f$-TVO VAEs associated with three well-known $f$-divergences (KL-divergence, Renyi's $\alpha$-divergence with $\alpha = 3$, and $\chi$-divergence with $n = 2$). We also try a customized divergence which is defined by its dual function $f_{c1}^*(t) = \log^2 t + \log t$.

We use the left Riemannian approximation with 5 partitions, i.e. $f$-TVO$^L(5)$ in our setting. As shown in Table 2, the proposed $f$-TVOs outperforms the corresponding baselines in most cases.

**Effects of numbers of partitions.** Here, we study the effects of different numbers of partitions on Bayesian variational autoencoder in our experiments. We try partition numbers $K = 1, 2, 5, 10$ in the $f$-TVO, respectively. As shown in Fig. 3, we can see that even partition number is $K = 2$ can improve the baseline performance.

**Effects of $f$-TVO variants.** We also study the effects of $f$-TVO variants defined in Sec. 4. We try all 3 variants $f$-TVO$^L$,$f$-TVO$^R$ and $f$-TVO$^Z$ on Bayesian variational autoencoder in our experiments. As shown in Fig. 4, three $f$-TVO variants achieve similar performance.

### 5.2 BAYESIAN NEURAL NETWORKS

Our experiments of Bayesian neural network are performed on regression tasks. We generally follow the experiments settings as in (Wan et al., 2020). In particular, We use twelve datasets that are collected from the UCI dataset repository, where each dataset is randomly split into 90%/10% for training and testing. The model is a single-layer neural network with 50 hidden units (ReLUs). We use $\theta \sim \mathcal{N}(\theta; 0, I)$ as a Gaussian prior of the network weights and Gaussian approximation to the true posterior.

In the experiments, we try different $f$-TVOs, including corresponds to KL TVO, $\alpha$-divergence TVO, where $\alpha = 3$, $\chi$-square divergence TVO. Following (Wan et al., 2020), we also define a custom divergence, which is defined by its dual function $f_{c2} = \tilde{f}^*(t) - \tilde{f}^*(1)$, where $\tilde{f}^*(t) =$

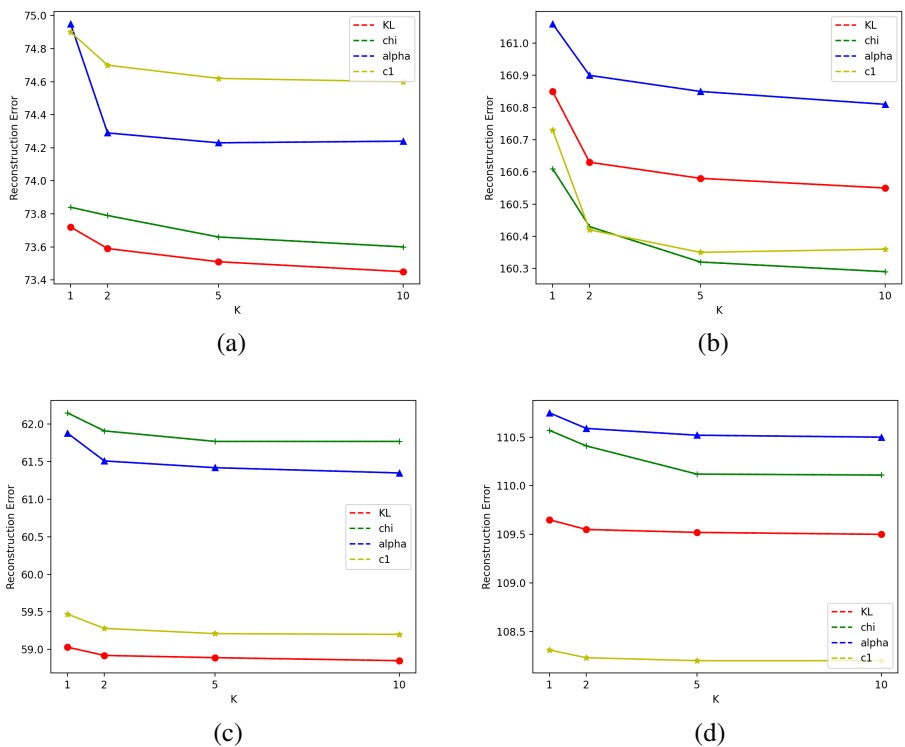

Figure 3: Effect of partition numbers. We evaluate $f$-TVO with partition numbers [1,2,5,10] on datasets (a)Caltech (b) Frey Face (c) MNIST (d) Omniglot.

| Dataset | KL-VI | KL-TVO | $\chi$-VI | $\chi$-TVO | $\alpha$-VI | $\alpha$-TVO | $f_{c2}$-VI | $f_{c2}$-TVO |
|---|---|---|---|---|---|---|---|---|
| Airfoil | 2.18 | 2.07 | 2.36 | **1.96** | 2.43 | 2.14 | 2.35 | 2.20 |
| Aquatic | 1.17 | 1.11 | 1.25 | 1.12 | 1.16 | 1.11 | 1.14 | **1.05** |
| Boston | 2.88 | **2.69** | 2.99 | 2.80 | 2.89 | 2.73 | 2.86 | 2.73 |
| Building | 1.56 | **1.38** | 2.81 | 2.36 | 1.87 | 1.64 | 1.85 | 1.65 |
| CCPP | 4.19 | 3.96 | 4.26 | 3.96 | 4.20 | **3.86** | 4.35 | 4.08 |
| Concrete | 5.40 | 5.38 | 3.51 | **3.37** | 5.40 | 5.24 | 5.26 | 5.11 |
| Fish Toxicity | 0.95 | 0.88 | 0.92 | 0.89 | 0.91 | 0.86 | 0.89 | **0.85** |
| Protein | 1.96 | 1.91 | 2.46 | 2.24 | 1.85 | **1.74** | 2.00 | 1.89 |
| Real Estate | 7.53 | 7.39 | 7.51 | **7.33** | 7.50 | 7.40 | 7.62 | 7.45 |
| Stock | 3.95 | 3.78 | 3.93 | **3.74** | 3.99 | 3.84 | 3.95 | 3.81 |
| Wine | .658 | .648 | .647 | .639 | .643 | **.635** | .665 | .649 |
| Yacht | 0.79 | **0.76** | 1.23 | 1.23 | 1.05 | 0.95 | 1.22 | 1.18 |

Table 3: Test RMSE of Bayesian Neural Network for $f$-VIs and $f$-TVOs. Lower is better. In most cases, $f$-TVOs outperform corresponding $f$-VIs.

$-1/6 \cdot (\log t + t_0)^3 - 1/2 \cdot (\log t + t_0) - 1$. We use the left Riemannian approximation with 5 partitions, i.e. $f$-TVO$^L(5)$ in our setting. As shown in Table 3, the proposed $f$-TVOs outperform the corresponding baselines.

## 6 CONCLUSION

In this paper, we have proposed a general $f$-TVO framework that is a tighter evidence bounds than $f$-VI. $f$-TVO unifies all differential $f$-divergence into TVO from a $\chi$-geometry perspective. By connecting $f$-divergence with $\chi$-geometry, $f$-TVO is the Riemanian sum of thermodynamic

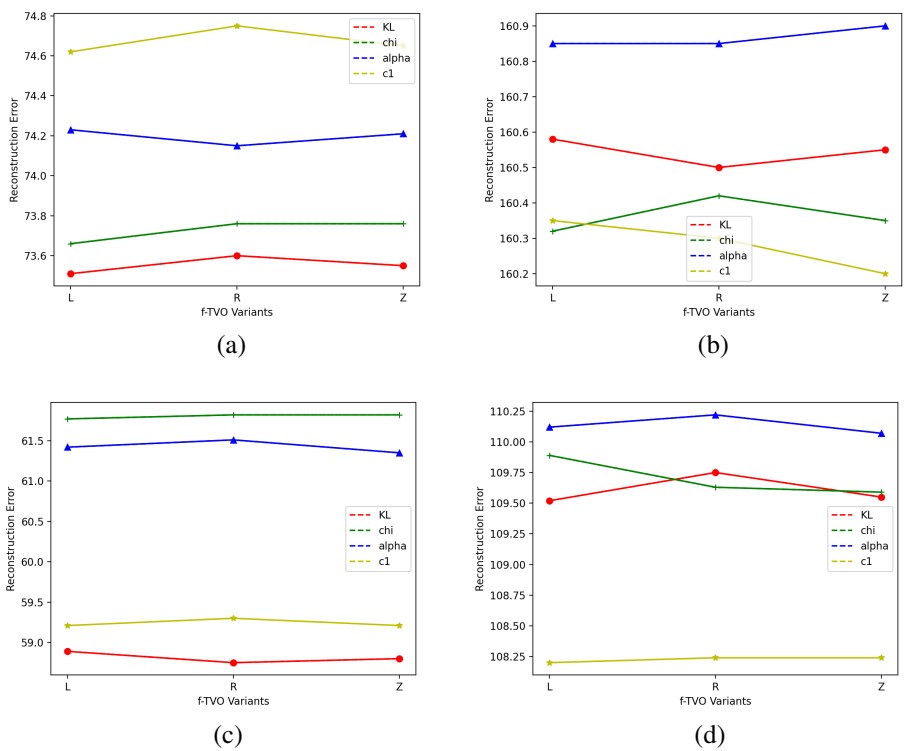

Figure 4: Effect of $f$-TVO variants. We evaluate $f$-TVO with all 3 variants: left riemannian approximation, right Riemannian approximation, "zig-zag" Riemannian approximation, on datasets (a)Caltech (b) Frey Face (c) MNIST (d) Omniglot.

| Dataset | KL-VI | KL-TVO | $\chi$-VI | $\chi$-TVO | $\alpha$-VI | $\alpha$-TVO | $f_{c2}$-VI | $f_{c2}$-TVO |
|---|---|---|---|---|---|---|---|---|
| Airfoil | 2.22 | 2.14 | 2.30 | **2.02** | 2.40 | 2.21 | 2.29 | 2.16 |
| Aquatic | 1.57 | 1.52 | 1.59 | 1.47 | 1.54 | 1.48 | 1.54 | **1.39** |
| Boston | 2.49 | **2.32** | 2.54 | 2.40 | 2.48 | 2.36 | 2.49 | 2.38 |
| Building | 6.90 | 6.71 | 6.85 | 6.75 | 6.79 | **6.56** | 6.74 | 6.58 |
| CCPP | 2.84 | 2.65 | 2.93 | 2.76 | 2.84 | **2.57** | 2.99 | 2.80 |
| Concrete | 3.23 | 3.02 | 2.75 | **2.51** | 3.12 | 2.87 | 3.10 | 2.83 |
| Fish Toxicity | 1.35 | 1.36 | 1.27 | **1.25** | 1.31 | 1.26 | 1.32 | 1.27 |
| Protein | 2.12 | 1.91 | 2.11 | 1.97 | 2.14 | **1.90** | 2.25 | 2.06 |
| Real Estate | 3.55 | 3.40 | 3.69 | 3.45 | 3.59 | **3.38** | 3.65 | 3.51 |
| Stock | -1.08 | -1.12 | -1.08 | -1.06 | -1.10 | -1.08 | -1.14 | **-1.18** |
| Wine | .975 | .974 | .970 | .966 | .971 | **.965** | .978 | .970 |
| Yacht | 1.72 | **1.68** | 1.88 | 1.82 | 1.87 | 1.87 | 2.02 | 1.95 |

Table 4: Test negative log-likelihood of Bayesian Neural Network for $f$-VIs and $f$-TVOs. Lower is better. In most cases, $f$-TVOs outperform corresponding $f$-VIs.

integration along the geodesic between $q(z|x)$ and $p(x, z)$ under $\chi$-geometry. Empirical experiments of some instances of $f$-TVO on the popular benchmarks show the superior performance than state-of-the art results, which imply the flexibility and effectiveness of $f$-TVO. Future work on $f$-TVO may include more efficient $f$-TVO optimization methods, and more explorations on the properties of $\chi$-geometry.

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
