# OpenReview forum: "$f$-Divergence Thermodynamic Variational Objective: a Deformed Geometry Perspective"
_ICLR.cc/2022/Conference — ICLR 2022 Submitted_

### Official Review · Reviewer_QFxg · 2021-10-30

**Correctness:** 2
**Technical Novelty And Significance:** 3
**Empirical Novelty And Significance:** 2
**Recommendation:** 3
**Confidence:** 4

**Main Review:**

The generalized $\chi$ exponential family part is interesting and new to me. But it seems inconsistency in the definition, the authors need to make careful and clear explanations and check every detail before submission.


Weakness:
- The definition of f-divergence is different from the commonly used one, $D_f(p||q) = \int q f(p/q) dz$.

- Equation 8 does not hold when $\log_{\chi}(u) = \log u$, $\chi(v)=1/v$, there is an additional constant 1.
- What is the domain of the function $\chi$?
- Convex differentiable generation function f with $f(0)=1$ or $f(1)=0$?
- How to guarantee the existence of the function \chi for any generation function f? For example, when $f^*(u)=u^2-1$ is convex differentiable and $f^*(1)=0$. When u is larger than 1, $f^*(u)$ is positive while $-\int_1^u \chi(v)dv$ is negative based on the condition on function $\chi$.
- Equation 11 $D_f(p||q) = -\int p(z) \log_\chi (\frac{p(z)}{q(z)})dz$.
- Can you provide the derivation of equation 17? I do not think this is a direct result.
- I did not find the monotonicity of $S_\beta$ in the paper. One key point in TVO is the integrand is monotonically increasing, thus the $\log$ probability can be estimated by the Riemann sum.

typo:
- page 2: $f$-divergence is a measures --> measure
- page 2: $f$-TVO in contributions should use the same font as the other part of the paper.
- page 4: we can define a exponential --> an exponential
-page 4: $x$ should be $\textbf{x}$ in definition 3.



**Summary Of The Paper:**

This paper proposed an $f$-divergence TVO, which includes some existing works e.g., RVB, CUBO, ELBO into a unified framework. This paper's main idea is to transform $f$-divergence into a generalized $\chi$-exponenetial family and integral TVO along the $\chi$-path. The paper provides some theoretical analysis and the optimization methods of the suggested framework and supports the proposed $f$-TVO with numerical results.

**Summary Of The Review:**

In all, the theoretical part is intriguing but not convincing enough. There are many typos and inconsistencies in the notations and computations, I expect the authors to provide some supplementary materials to support the theory, including all the mathematical derivations.

---

> ### Author Response · Authors · 2021-11-22
> **Thank you for your comments. The followings are our answer.**
>
> Q1: Equation 8 does not hold when $\log_{\chi}(u) = \log u$, there is an additional constant 1
>
> A1: This is not correct. There is no constant 1, as $\log 1$ = 0
>
>
> Q2. What is the domain of the function $\chi$?
>
> A2. We mentioned about the equation 8 "a positive non-increasing function $\chi$". For the domain of $\chi$, it depends on the particular $\chi$, for example, the domain $\log u$ is $(0, +\infty)$.
>
>
> Q3: Convex differentiable generation function f with  f(0) = 1 or f(1)=0?
>
> A3: It's a type. It should be f(1)=0. Thanks for the heads up.
>
>
> Q4: How to guarantee the existence of the function $\chi$ ?
>
> A4: There is a type in Eq 10. There should be no negative sign in Eq 10. We will modify it in our paper.
>
>
> Q5: Can you provide the derivation of equation 17?
>
> A5: Yes. We will add derivation of Eq 17 in our paper.

---

### Official Review · Reviewer_BE5D · 2021-11-01

**Correctness:** 3
**Technical Novelty And Significance:** 3
**Empirical Novelty And Significance:** 1
**Recommendation:** 3
**Confidence:** 4

**Main Review:**

The authors provided an interesting theoretical extension of TVO to more general f-divergence families. However, this extension seems not surprising but straightforward. The critical issue of this work is that the authors did not present the motivation for the extension. I expected that numerical experiments support the motivation of extending the f-VI to f-TVO. But the presented numerical experiments are too weak, and I could not find the usefulness of the proposed method. It seems that standard KL-TVO still works very well. I strongly recommend adding additional experiments to support the usefulness of the f-TVO.

Pros
-  Combine two important concepts in VI, TVO and f-VI.

Cons
- Numerical experiments are too weak to support the usefulness of the proposed method. I could not find the practical benefit of the generalization of TVO to f-VI.
- Experiments are only done for over-parametrized models. But there is no explanation why f-TVO is recommended to use for over-parametrized models.
- The details of the experimental settings are lacking or insufficient to reproduce the results.

Comments and Questions:
- Please provide the proof for the statement before Eq10
- No proof of Eq17
- Below Eq22, $q(z|x)=p(\epsilon)$ seems strange, should be $p(g(\epsilon))$ ?
- Could not follow the discussion below Eq24. Please explicitly write how the authors used the logsumexp trick here. Moreover, if $R$ and $L$ are negative, $\ln R$ and $\ln L$ cannot be defined.

- In the numerical experiments, the authors introduced the customized f-divergence, called $f_{c2}$. I know that they were introduced in the previous work [29], but the authors should explain why it is used in numerical experiments and what distinguished properties they have compared to other f-divergences so that non-experts can understand them.
- What is the reason for introducing the zig-zag integral? In VAE experiments, it did not show significant improvements.
- In VAE experiments, I could not find the network architectures. Also, I would like to know the performance in the more meaningful measures, such as the test log-likelihood. Moreover, the authors should compare the results with IWAE, since IWAE also uses the multi-sample bound.
- Based on all the numerical experiments, it seems that KL-TVO seems to show the best performance on average and could not confirm why we use f-TVO.

- All the experiments are done in over-parametrized models. Why?
- I would like to see results of simple models, for example, GLM, mixture models, or Gaussian processes, in which we can estimate true posterior distributions.  We can understand the solution of f-TVO more intuitively in those models than in over parametrized models.

**Summary Of The Paper:**

This paper proposed new variational inference that combines f-divergence variational inference and the thermodynamic variational objective. The authors introduced several new concepts of exponential families to extend TVO. Finally, the authors provided the estimator of the gradient of the objective function based on the reparametrization trick.

**Summary Of The Review:**

The extension of TVO to f-TVO is an interesting direction, but I think the experimental results are too weak to support the usefulness of the proposed method. Moreover, the description of the experimental settings is limited to reproduce the results. I recommend improving writing and experimentation.

---

> ### Author Response · Authors · 2021-11-22
> **Thank you for your comments. The followings are our answer.**
>
> Q1: The details of the experimental settings are lacking or insufficient to reproduce the results.
>
> A1: We mention in the paper we follow the prior work for the network architecture and training strategy, We will add more details in our paper.
>
>
> Q2: Please provide the proof for the statement before Eq10 and Eq17
>
> A2: We will add the detailed proofs
>
>
> Q3: Please explicitly write how the authors used the logsumexp trick here. Moreover, if  L and R are negative, L and R cannot be defined.
>
> A3: Actually L and R will not be negative.  We will add more details for some concrete examples.
>
>
> Q4:  why it is used in numerical experiments  for $f_{c2}$
>
> A4: We will explain in the paper that this divergence cannot be in the $\alpha$-divergence framework, while most existing divergences can be in the $\alpha$-divergence framework.
>
>
> Q5: What is the reason for introducing the zig-zag integral?
>
> A5. As shown in Table 3 and 4, the zig-zag integral performs the best in some settings.
>
>
> Q6: All the experiments are done in over-parametrized models. Why?
>
> A6: We focus our experiment on Bayesian networks and VAE. Both rely on deep neural networks that are over-parameterized. We may add more experiments on some shallow neural network architectures.

---

### Official Review · Reviewer_Qe9J · 2021-11-02

**Correctness:** 3
**Technical Novelty And Significance:** 4
**Empirical Novelty And Significance:** 2
**Recommendation:** 5
**Confidence:** 3

**Main Review:**

Overall the paper is well written, however, it lacks some more motivation (and insights) about the extension enabled by deformed geometry. Moreover, the experiments should be improved.

strengths:
 - novel extensions of TVO towards f-divergences
 - solid theoretical work

weakness:
- experiments should be improved and better discussed
- lack of discussion (and comparison) that connects to related TVO work [1]


- It is not immediately clear what are the clear benefits of f-TVO, wrt other objectives that can be defined within TVO (ELBO, EUBO, Renyi, CUBO etc) [1].
- The related work section can be improved, discussing work on TVO that already have shown improvements (such as refined partitioning in [1]).
- The experiments can also be improved, showing standard (from multiple runs) to begin with, and providing more discussion on the results. As of now, here isn't some significant or practical benefits between f-TVO variants and f-VI. The authors also introduce two additional divergences (c1 and c2) which are not motivated and discussed in more detail.

Other comments:
- Table 1: What is H_alpha ()? (Hellinger?)
- How were the number of partitions (K=5) and the f-TVO variant (left) chosen for the experiments? It seems that the performance can slightly vary wrt. the choice of divergence and dataset? Is there any intuition behind the choice?

[1] Brekelmans et al (2020). All in the exponential family: Bregman duality in thermodynamic variational inference.

**Summary Of The Paper:**

The paper proposes a novel f-divergence Thermodynamic Variational Objective (f-TVO) framework for VI, that extends the TVO towards, a more general, family of f-divergences. The authors propose to use a $\chi$-deformed exponential distribution, which casts the f-TVO objective as integral along the $\chi$-path between p(x,z) and q(z|x) (rather than the geometric path in TVOs) under $\chi$-geometry. The authors propose different variants of f-TVO, that vary between the type of the f-divergance used, as well as how this integral is approximated (K-partitioned ) left-Riemann sum (related to ELBO), right-Riemann sum (related to EUBO) or a 'zig-zag' that alternates between the two. Besides theoretical justifications, results from two sets of experiments show that, in general, the proposed f-TVO perform comparable to or slightly better than the f-VI counterparts, but without clear conclusion wrt the choice of the f-divergence.

**Summary Of The Review:**

Overall the paper is well written and (mostly) easy to follow. The ideas and contributions are novel and solid: connecting f-divergances with TVO. However, it lacks some more motivation (and insights) about the connection with deformed geometry. The empirical results are not convincing, since the reported performances, while similar across methods, do not include standard error nor more discussion, which would better highlight the robustness and benefits of the proposed f-TVOs.

----
Post-rebuttal update:
Many thanks to the authors for addressing some of my concerns. However, I still think that the experimental setup should be improved and some design choices (eg. wrt the custom divergences) better motivated and evaluated. With this in mind, as well as reading the other reviews (and responses), my initial recommendation will remain the same.

---

> ### Author Response · Authors · 2021-11-22
> **Thank you for your comments. The followings are our answer.**
>
> Q1: lack of discussion (and comparison) that connects to related TVO work [1]
> A1: This is not correct. We discuss the two difference compared to the TVO in abstract. 1) we generalize TVO by replacing Kullback–Leibler (KL) divergence with arbitary differeitiable divergence. 2) f-TVO approximates dual function of model evidence  rather than the log model evidence.
>
> Q2: It is not immediately clear what are the clear benefits of f-TVO, wrt other objectives that can be defined within TVO
> A2: We mention our contribution in the introduction. In theory, we enrich the TVO by proposing a unified f-TVO framework. In experiment, the proposed f-TVO is a better optimization surrogate than the corresponding f-VI.
>
> Q3: The related work section can be improved, discussing work on TVO that already have shown improvements
> A3: Sure. We will add an additional related work section that summarizes TVO that already have shown improvements  and f-divergnce.
>
> Q4 : Table 1: What is H_alpha ()? (Hellinger?)
> A4 :  As shown in the caption of Table 1, H_alpha is Hellinger.

---

### Official Review · Reviewer_6bLb · 2021-11-04

**Correctness:** 3
**Technical Novelty And Significance:** 2
**Empirical Novelty And Significance:** 3
**Recommendation:** 3
**Confidence:** 4

**Main Review:**

The general idea does not seem surprisingly novel. Nevertheless, developing the objective and its calculation are technically nontrivial and worth a contribution. Still, I think the current paper does not seem to implement the idea in a comprehensive and compelling way.

Presentation issues:
* The main technique content introduces what the bound is, how is it developed and how to do the calculation. But I expect more comprehensive (unnecessarily deep and involved) technical investigations that help readers grasp a general understanding of the developed bound. For example, why is it better (e.g., how is it tighter if it is) than existing bounds, particularly f-VI and TVO? When is the bound tight? How does it relate to other bounds? These issues are also related to the motivation for developing this bound and explaining why the new bound achieves a better performance.
* Eq. (17) seems to be the central technical contribution in this paper, but its derivation is missing. I took some effort and verified the equation myself by the facts listed in the paper, but I still think missing the derivation is a problem. The derivation is even more important than the proof of Theorem 2.
* The potential energy function $U_\beta(z)$ is introduced but not used anywhere.
* I do not quite understand why Theorem 2 is explained as that "$\tilde\pi_\beta(z)$ is on the log $\chi$ geodesic between $\tilde\pi_0(z)$ and $\tilde\pi_1(z)$". To me, it just shows how $\tilde\pi_\beta(z)$ interpolates between the two ends. If geodesic is mentioned, then (1) conceptually, specify the metric or Riemannian structure that defines the geodesic, and (2) why consider the geodesic, i.e. how the properties of geodesic relate to our desideratum.
* Some claims are expecting proof or reference. For example, Eq. (6) (there is no reference in Sec. 2.2), the statement above Eq. (10), and Eq. (17).
* Above Eq. (7), $Z_\beta$ is a normalizing constant over $(x,z)$ or $z$?
* There are some grammar and spelling typos ("differeitiable" and "able to integral" in abstract, "knowledges" in introduction, "a exponential family", "two unnormalized distribution" and "pick the a list" in Sec. 3.2).

Technical problems:
* In Eq. (21), what is $q(z)$? To make the equality, it should be $q(z|x)$ right? If yes, then why does the reparameterization map $g_\phi$ does not also take $x$ as an input?
* Eq. (22) seems problematic. "Defining $q(z|x) = p(\epsilon)$" may break the equality. Instead, $p(\epsilon)$ in Eq. (22) should be $q(z=g_\phi(\epsilon) | x)$, which differs from $p(\epsilon)$ by a factor $1 / \sqrt{|\Sigma|}$ (this can also be derived by the rule of change of variables), which cannot be ignored since it depends on $\phi$ (and $x$).
* The calculation of $w_n$ also seems problematic. In the equation below Eq. (23), the $1/N$ factor seems missing in the argument of the firxt $\chi$. Also, the summation estimates $\int \tilde\pi_\beta(z) q(z|x) \\, \mathrm{d}z$, but not $Z_\beta$ which is $\int \tilde\pi_\beta(z) \\, \mathrm{d}z$.
* How is $\tilde\pi_\beta(z)$ evaluated? It requires calculating the $\exp_\chi$ map, which by definition is rather implicit. So is it evaluated by iteratively solving for the inverse of $\log_\chi$, or is known in closed-form? If the latter, how is it known?

**Summary Of The Paper:**

The paper presents a new bound as the objective for variational inference. The bound combines the recent progress of thermodynamic variational object which gives a tighter bound than the conventional ELBO, and the f-divergence which induces more possible distribution metrics. Experiments show its better performance on some Bayesian inference tasks.

**Summary Of The Review:**

Up to my understanding, there are some technical flaws and the paper does not investigate the proposed bound comprehensively (particularly, the motivation is not supported and the advantage is not clear).

---

> ### Author Response · Authors · 2021-11-22
> **Thank you for your comments. The followings are our answer.**
>
> Q1: Eq. (17) seems to be the central technical contribution in this paper, but its derivation is missing. The potential energy function
>  is introduced but not used anywhere.
>
> A1: Thank you for your comment. We will add a detailed derivation of Eq(17) in our paper. The potential energy function will also be included in the derivation.
>
>
> Q2: I do not quite understand why Theorem 2 is explained about the log geodesic
>
> A2: We will explain it in the paper that the geodesic distance between $\pi_{0}$ and $\pi_{1}$ is a linear combination of the geodesic distance between $\pi_{0}$ and $\pi_{\beta}$ and the geodesic distance between $\pi_{\beta}$ and $\pi_{1}$.
>
>
> Q3:Above Eq. (7),  the normalizing constant is over (x,z) or z ?
>
> A3: We will explain the normalizing constant is over z in the paper.
>
>
> Q4: Eq. (22) seems problematic. $p(\epsilon)$   in Eq. (22) should be $q(z = g_{\phi(\epsilon)}|x)$
>
> A4: Thank you for your comment. We will modify it in our paper.
>
>
> Q5: How is $\tilde{\pi}_{\beta}$ evaluated?
>
> A5: We already explain our computation in Section 3.3. We will add more examples to the paper.

---

### Decision · Program_Chairs · 2022-01-20

**Decision:**

Reject

**Comment:**

The four reviewers believed the paper was below threshold for acceptance to ICLR. They raised concerns with the experimental evaluation and thought that the paper could benefit from another edit to help with the clarity.